# Upper bound on the biological effects of 50/60 Hz magnetic fields mediated by radical pairs

PJ Hore*

Department of Chemistry, Physical & Theoretical Chemistry Laboratory, University of Oxford, Oxford, United Kingdom

**Abstract** Prolonged exposure to weak (~1 µT) extremely-low-frequency (ELF, 50/60 Hz) magnetic fields has been associated with an increased risk of childhood leukaemia. One of the few biophysical mechanisms that might account for this link involves short-lived chemical reaction intermediates known as radical pairs. In this report, we use spin dynamics simulations to derive an upper bound of 10 parts per million on the effect of a 1 µT ELF magnetic field on the yield of a radical pair reaction. By comparing this figure with the corresponding effects of changes in the strength of the Earth's magnetic field, we conclude that if exposure to such weak 50/60 Hz magnetic fields has any effect on human biology, and results from a radical pair mechanism, then the risk should be no greater than travelling a few kilometres towards or away from the geomagnetic north or south pole.

DOI: https://doi.org/10.7554/eLife.44179.001

## Introduction

Residents in developed countries are almost continuously exposed to the 50 or 60 Hz electromagnetic fields generated by electrical appliances and power transmission lines. Although these fields are normally very weak in residential environments (electric component <100 V m$^{-1}$, magnetic component <1 µT (*Swanson and Kheifets, 2006*)), these extremely-low-frequency (ELF) fields have been mooted as a potential health hazard (*Crumpton, 2005*; *Crumpton and Collins, 2004*; *Brocklehurst and McLauchlan, 1996*). In relation to childhood leukaemia, ELF magnetic fields have been classified as 'possibly carcinogenic to humans' (*Ahlbom et al., 2000*; *Greenland et al., 2000*; *IARC, 2002*; *Kheifets et al., 2010*). Although there is scant evidence from animal experiments to support a causative link (*IARC, 2002*; *WHO, 2007*), numerous in vitro investigations have reported biological effects of ELF fields (reviewed in *Juutilainen et al., 2006*; *Mattsson and Simkó, 2014*; *Juutilainen et al., 2018*). Not many of these observations have been independently replicated (*Lacy-Hulbert et al., 1998*). To explain any genuine effects, there must exist biophysical mechanisms by which weak ELF magnetic fields interact with molecules in living organisms. More than 10 years ago, Swanson and Kheifets (*Swanson and Kheifets, 2006*) and Adair (*Adair, 1991*; *Adair, 1999*; *Adair, 2000*) assessed a range of potential mechanisms and concluded that none was likely to have biological consequences at magnetic flux densities of order 1 µT.

One of the mechanisms considered was the radical pair mechanism (*Brocklehurst and McLauchlan, 1996*). Unlike some of the others, this is unquestionably a genuine phenomenon supported by hundreds of laboratory studies of (mostly non-biological) chemical reactions in magnetic fields ranging from ~1 mT up to several Tesla (*Steiner and Ulrich, 1989*; *Brocklehurst, 2002*; *Rodgers, 2009*; *Jones, 2016*; *Hore, 2012*; *Scaiano et al., 1994a*). Radical pairs are short-lived reaction intermediates, typically formed in highly non-equilibrium electron-spin states. Their coherent spin dynamics, combined with spin-dependent reactivity, allow magnetic interactions a million times

*For correspondence:
peter.hore@chem.ox.ac.uk

**Competing interests:** The author declares that no competing interests exist.

smaller than the thermal energy, $k_B T$, to have measurable, reproducible and predictable effects on chemical reaction yields. In the last few years, interest in the biological significance of <1 mT static magnetic fields has been kindled by new insights into the biophysical mechanism of the avian magnetic compass sense (*Mouritsen, 2018*; *Hore and Mouritsen, 2016*; *Rodgers and Hore, 2009*; *Nordmann et al., 2017*; *Schulten et al., 1978*). It now seems likely that migratory songbirds detect the direction of the geomagnetic field by means of photo-induced radical pairs generated within cryptochrome proteins located in their retinas (*Hore and Mouritsen, 2016*; *Ritz et al., 2000*; *Liedvogel and Mouritsen, 2010*). The growing support for this hypothesis includes an experimental demonstration that a radical pair reaction can be influenced by a static magnetic field of the same strength as the geomagnetic field (~50 µT) (*Maeda et al., 2008*). Whether this hypothesis proves to be correct or not, it is important to distinguish between functional responses, such as magnetoreception, which presumably have been optimised by evolutionary pressure, and fortuitous, non-functional magnetic effects which could in principle be damaging. Since humans seem to navigate without the benefit of in-built magnetic sensors, we focus here exclusively on the latter. However, one cannot exclude the possibility that other biological responses to the Earth's magnetic field, for example at the cellular level, could have been useful at an early stage in human evolution and might still persist.

Prompted by their putative role in magnetoreception, cryptochromes have become the focus of recent discussions of fortuitous radical pair effects in biology (*Bounds and Kuster, 2015*; *Lagroye et al., 2011*; *Vanderstraeten et al., 2015*; *Close, 2012*; *Close, 2014a*; *Close, 2014b*; *Landler and Keays, 2018*; *Krylov, 2017*; *Agliassa et al., 2018*). Juutilainen et al., for example, have proposed a hypothesis to explain the link between environmental ELF magnetic fields and childhood leukaemia ('magnetocarcinogenesis') (*Juutilainen et al., 2018*). Because cryptochromes are key components of the circadian clock (*Chaves et al., 2011*; *Kelleher et al., 2014*; *Hastings et al., 2014*), and circadian systems are closely coupled to the regulation of DNA damage responses and defence against reactive oxygen species, it is possible that magnetic field effects on radical reactions in cryptochromes could lead to circadian dysregulation, genomic instability and ultimately cancer (*Juutilainen et al., 2018*).

Our purpose here is to extend Adair's 1999 discussion (*Adair, 1999*) of radical pair magnetic field effects by modelling the spin dynamics of cryptochrome-based radical pairs using more advanced theoretical methods. The goal is to derive a realistic order-of-magnitude estimate of the largest likely effect of a 1 µT ELF field in the presence of the Earth's magnetic field. This upper bound is compared with the predicted effects of modest changes in the strength of the geomagnetic field such as would be experienced by travelling around the globe and with the small changes in body temperature that occur naturally in healthy humans.

## Methods

### Outline

Our aim is to determine the change in the yield of a radical pair reaction caused by a 1 µT ELF field superimposed on the geomagnetic field. We make no attempt to link this change to any specific biological process; rather we seek to estimate the maximum primary magnetic field effect under chemically and physically plausible conditions. Calculations are based on the [FAD•⁻ TrpH•⁺] radical pair that accounts for the observed effects of static magnetic fields on the photochemistry of purified cryptochromes (*Hore and Mouritsen, 2016*; *Maeda et al., 2012*; *Sheppard et al., 2017*; *Kattnig et al., 2016a*). This species is formed by the transfer of an electron from a tryptophan residue (TrpH) in the protein to the photo-excited, non-covalently bound, flavin adenine dinucleotide (FAD) chromophore, *Figure 1(a)*. We consider a simplified spin system comprising the two unpaired electron spins, one on each radical, coupled to three nitrogen nuclei ($^{14}$N, spin quantum number $I = 1$) chosen for their large isotropic hyperfine coupling constants: N5 and N10 in FAD•⁻ ($a = 523$ µT and 189 µT) and N1 in TrpH•⁺ ($a = 322$ µT), calculated using density functional theory (*Lee et al., 2014*). The radicals were assumed to have g-values equal to the free electron g-value, 2.0023. At the magnetic field strengths of interest here the difference in the Zeeman interactions of FAD•⁻ and TrpH•⁺ is entirely negligible. We exclude the anisotropic components of the hyperfine interactions which are only relevant when the radicals are immobilised and aligned, as in the case of a magnetic

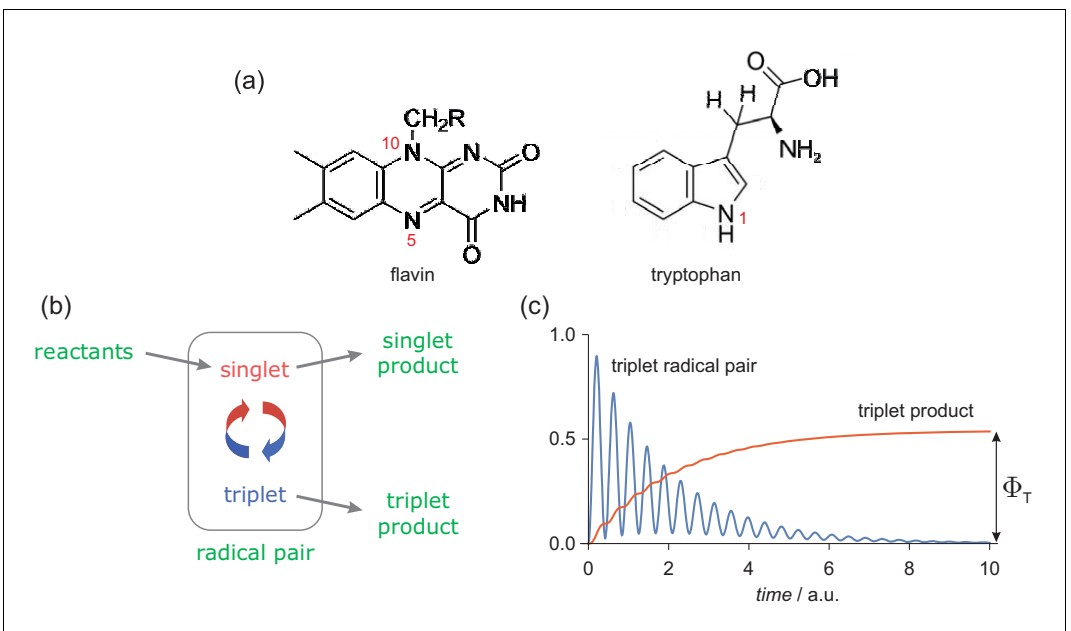

**Figure 1.** The radical pair mechanism. (**a**) Structures and atom numbers schemes for the FAD and TrpH molecules from which magnetically sensitive radical pairs are formed in purified cryptochromes. (**b**) Simple reaction scheme for a singlet-born radical pair able to react spin-selectively to form singlet and triplet reaction products. The red/blue arrows represent the coherent interconversion of the two forms of the radical pair. The reaction scheme is a simplified version of the cryptochrome photocycle (*Hore and Mouritsen, 2016*; *Maeda et al., 2012*). (**c**) Schematic dependence of the amounts of triplet radical pair and triplet product present as a function of time (in arbitrary units) after the formation of the radical pair in a singlet state. The triplet yield, $\Phi_T$, is the amount of triplet product formed once all radical pairs have reacted.

DOI: https://doi.org/10.7554/eLife.44179.002

compass sensor (*Schulten et al., 1978*). These hyperfine interactions condition the response of the radical pair to both static and ELF magnetic fields.

The chemistry of this radical pair was modelled in a simplified fashion by means of competing, spin-selective, singlet and triplet reaction channels (*Figure 1(b)*). For the present purposes, the precise nature of these reactions is immaterial but to avoid unnecessary abstraction we briefly describe the reaction steps in the context of a cryptochrome-based magnetic compass sensor (*Maeda et al., 2012*). Formed in a spin-correlated singlet state (total electron spin quantum number, $S = 0$), the radical pair coherently interconverts with the corresponding triplet state ($S = 1$) as a result of the internal hyperfine interactions and the Zeeman interactions of the electron spins with any external magnetic fields (*Hore and Mouritsen, 2016*). In cryptochrome, the singlet channel is a reverse electron transfer reaction that returns the charge-separated radical pair state to the diamagnetic ground state of the protein. The triplet channel is considered to produce the signalling state that initiates magnetic signal-transduction (*Hore and Mouritsen, 2016*). For simplicity, the two reaction channels were assigned identical first order rate constants, $k$ (the so-called 'exponential model' (*Timmel et al., 1998*)). Magnetic field effects arise from the competition between these two spin-conserving reactions together with the change in the extent and timing of the coherent singlet-triplet interconversion brought about by the external magnetic field(s). Although these calculations were performed for a highly simplified model of a radical pair state of cryptochrome, we will argue below (section titled 'Assumptions and approximations') that general conclusions can be drawn that are relevant to any chemically and physically realistic radical pair reaction.

Radical pairs can be influenced by magnetic interactions that are minuscule compared to $k_B T$ because they are formed in a spin-coherent state far removed from thermal equilibrium. To be sensitive to a weak static magnetic field, the coherence must persist for a time comparable to the period of the electron Larmor precession which, in a 50 µT magnetic field, is ~700 ns (*Rodgers and Hore, 2009*). Theoretical treatments often assume negligible spin relaxation during the radical pair lifetime; this is almost always unrealistic (*Hore and Mouritsen, 2016*; *Kattnig et al., 2016b*;

*Kattnig et al., 2016c*; *Worster et al., 2016*). The reality is that random molecular motions modulate the local magnetic fields experienced by the electron spins causing irreversible loss of spin coherence. Spin relaxation was included here by means of a phenomenological exponential decay towards the equilibrium state (25% singlet, 75% triplet), with a rate constant $r = 10^6 \, \text{s}^{-1}$, corresponding to a spin-coherence lifetime, $r^{-1} = 1 \, \mu\text{s}$. This choice of $r$ is discussed in 'Assumptions and approximations'.

Mutual exchange and dipolar interactions of the two unpaired electrons (*Efimova and Hore, 2008*) were ignored. The effects of external magnetic fields, both static and time-dependent, were quantified by calculating $\Phi_\text{T}$, the fraction of radical pairs that react via the triplet channel ($0 \leq \Phi_\text{T} \leq 1$, *Figure 1(c)*). We refer to this quantity as the triplet yield. The validity and implications of all the approximations and assumptions involved are discussed in 'Assumptions and approximations'.

## Calculation of $\Phi_\text{T}$

In the absence of spin relaxation and chemical reactions, the probability that a radical pair is in a triplet state a time $t$ after being created in a singlet state (compare equation (4) of *Timmel et al., 1998*) is

$$p'_\text{T}(t) = 1 - \frac{1}{M}\sum_m \sum_n |\langle m|\hat{P}^\text{S}|n\rangle|^2 \cos([\omega_m - \omega_n]t),$$

(1)

in which $\hbar\omega_i = \langle i|\hat{H}|i\rangle$, where $|i\rangle$ and $\hbar\omega_i$ are, respectively, the eigenstates and eigenvalues of the radical pair spin Hamiltonian, $\hat{H}$, which accounts for the hyperfine and Zeeman interactions of the radicals. $\hat{P}^\text{S}$ is the singlet projection operator and $M$ is the total number of nuclear spin configurations (*Timmel et al., 1998*). The oscillatory time-dependence of the triplet probability reflects the formation of the radical pair in a coherent, non-stationary, superposition state.

The spin Hamiltonian of the model [FAD$^{\bullet-}$ TrpH$^{\bullet+}$] radical pair was

$$\hat{H} = \omega(\hat{S}_{\text{A}z} + \hat{S}_{\text{B}z}) + a_1\hat{\mathbf{S}}_\text{A}.\hat{\mathbf{I}}_1 + a_2\hat{\mathbf{S}}_\text{A}.\hat{\mathbf{I}}_2 + a_3\hat{\mathbf{S}}_\text{B}.\hat{\mathbf{I}}_3$$

(2)

with the following spin angular momentum operators: $\hat{\mathbf{S}}_\text{A}$ and $\hat{S}_{\text{A}z}$, electron in FAD$^{\bullet-}$; $\hat{\mathbf{S}}_\text{B}$ and $\hat{S}_{\text{B}z}$, electron in TrpH$^{\bullet+}$ (or Z$^\bullet$, see later); $\hat{\mathbf{I}}_1$ and $\hat{\mathbf{I}}_2$ nitrogens N5 and N10 in FAD$^{\bullet-}$; $\hat{\mathbf{I}}_3$, nitrogen N1 in TrpH$^{\bullet+}$. $\omega = \gamma_e B$ is the electron Larmor frequency in a static magnetic field of strength $B$ and $a_n$ ($n = 1 - 3$) are the hyperfine coupling constants.

Spin relaxation was introduced phenomenologically (compare equation (19) of *Bagryansky et al., 2007*) as:

$$p_\text{T}(t) = \frac{3}{4} + \left(p'_\text{T}(t) - \frac{3}{4}\right)\exp(-rt),$$

(3)

where $r$ is the relaxation rate constant.

Following (*Timmel et al. (1998)*, the chemical fate of the radical pair was modelled by means of separate spin-selective reactions of the singlet and triplet pairs (*Figure 1(b)*). To simplify the calculation, these two processes were assigned identical first order rate constants, $k$. The ultimate yield of the product formed spin-selectively from the triplet radical pair state at a time $t \gg k^{-1}$ is therefore:

$$\Phi_\text{T} = k\int_0^\infty p_\text{T}(t)\text{e}^{-kt}\text{d}t$$
$$= \frac{3}{4} + \frac{k}{4(k+r)} - \frac{1}{M}\sum_m \sum_n |\langle m|\hat{P}^\text{S}|n\rangle|^2 \frac{k(k+r)}{(k+r)^2 + (\omega_m - \omega_n)^2}.$$

(4)

*Equation (4)* was used to calculate $\Phi_\text{T}$ and hence *mfe*$_\text{ELF}$ and *mfe*$_\text{GMF}$ (see below).

## Calculation of *mfe*$_\text{ELF}$ and *mfe*$_\text{GMF}$

The same model radical pair was modelled with a weak ELF field superimposed on the geomagnetic field (GMF, $B_0 = 50 \, \mu\text{T}$). Epidemiological studies of childhood leukaemia (*Ahlbom et al., 2000*; *Greenland et al., 2000*; *Kheifets et al., 2010*) found a two-fold increase in risk for chronic exposure to ELF magnetic fields of root-mean-square magnetic flux density $\geq 0.4 \, \mu\text{T}$ with an average strength

of ~0.7 μT. The latter corresponds to a peak intensity of $0.7 \times \sqrt{2} \approx 1.0$ μT. We therefore take the ELF field to have peak magnetic flux density $B_1 = 1.0$ μT and frequency 50 Hz. This field is assumed to be linearly polarised and aligned parallel to the GMF such that the total field experienced by the radicals varies between 49 and 51 μT. Any other relative orientation of the two fields would result in a smaller range of total field strengths and smaller ELF effects (see 'Assumptions and approximations').

Generally requiring lifetimes between 100 ns and 10 μs for a significant low field effect (see 'Static magnetic fields'), magnetically sensitive radical pairs have a fleeting existence compared to the 20 ms period of a 50 Hz electromagnetic wave. Consequently the ELF field can be treated as effectively static during the lifetime of a radical pair (*Scaiano et al., 1994b*). We suppose that radical pairs are formed continuously, for example by photo-excitation of cryptochrome. Each radical pair therefore experiences a different, effectively static, magnetic field whose intensity, $B$, depends on the phase of the ELF field, $\alpha$, which barely changes during the lifetime of the pair,

$$B = B_0 + B_1 \cos\alpha, \tag{5}$$

with $\alpha$ randomly distributed in the range $(0, \pi)$. The net effect of the ELF field on an ensemble of independently created radical pairs is an average over $\alpha$,

$$\overline{\Phi_{\mathrm{T}}(B_0, B_1)} = \frac{1}{\pi} \int_0^\pi \Phi_{\mathrm{T}}(B)\mathrm{d}\alpha. \tag{6}$$

We define the magnetic field effect, $mfe_{\mathrm{ELF}}$, as the relative difference between $\overline{\Phi_{\mathrm{T}}(B_0, B_1)}$ and $\Phi_{\mathrm{T}}(B_0)$, the triplet yield in the absence of the ELF field:

$$mfe_{\mathrm{ELF}} = \frac{\overline{\Phi_{\mathrm{T}}(B_0, B_1)} - \Phi_{\mathrm{T}}(B_0)}{\Phi_{\mathrm{T}}(B_0)}. \tag{7}$$

$mfe_{\mathrm{ELF}}$ may be evaluated by using $B_1 << B_0$ to expand $\Phi_{\mathrm{T}}(B)$ as a Taylor series around $B = B_0$, to second order in $B - B_0$:

$$\Phi_{\mathrm{T}}(B) \approx \Phi_{\mathrm{T}}(B_0) + (B - B_0)\Phi_{\mathrm{T}}^{(1)}(B_0) + \frac{1}{2}(B - B_0)^2 \Phi_{\mathrm{T}}^{(2)}(B_0), \tag{8}$$

where $\Phi_{\mathrm{T}}^{(n)}(B_0)$ is the $n$-th derivative of $\Phi_{\mathrm{T}}(B)$ evaluated at $B = B_0$. Combining *Equations (5), (6) and (8)* gives the average triplet yield as a sum of zero and second order terms in $B_1$:

$$\overline{\Phi_{\mathrm{T}}(B_0, B_1)} \approx \Phi_{\mathrm{T}}(B_0) + \frac{1}{4}B_1^2 \Phi_{\mathrm{T}}^{(2)}(B_0). \tag{9}$$

Combining *Equations (7) and (9)* gives:

$$mfe_{\mathrm{ELF}} \approx \frac{1}{4}B_1^2 \frac{\Phi_{\mathrm{T}}^{(2)}(B_0)}{\Phi_{\mathrm{T}}(B_0)}. \tag{10}$$

When $B_1 << B_0$, the magnetic field effect, $mfe_{\mathrm{ELF}}$, is thus proportional to the second derivative (i.e. the curvature) of $\Phi_{\mathrm{T}}(B)$ at $B = B_0$ and to the square of the intensity of the ELF field (*Adair, 1999*; *Adair, 1994*).

By analogy with *Equation (7)*, we also define a geomagnetic field effect, $mfe_{\mathrm{GMF}}$, as the fractional change in the triplet yield in the absence of an ELF field, when the static magnetic field is changed from $B_0$ to $B_0 + \Delta B_0$ where $|\Delta B_0| \ll B_0$:

$$mfe_{\mathrm{GMF}} = \frac{\Phi_{\mathrm{T}}(B_0 + \Delta B_0) - \Phi_{\mathrm{T}}(B_0)}{\Phi_{\mathrm{T}}(B_0)}. \tag{11}$$

Expansion of $\Phi_{\mathrm{T}}(B)$ as a Taylor series around $B = B_0$, to first order in $\Delta B_0$, gives

$$mfe_{\mathrm{GMF}} \approx \Delta B_0 \frac{\Phi_{\mathrm{T}}^{(1)}(B_0)}{\Phi_{\mathrm{T}}(B_0)}, \tag{12}$$

which should be compared with *Equation (10)*. In contrast to $mfe_{\mathrm{ELF}}$, $mfe_{\mathrm{GMF}}$ depends on the *first* derivative (i.e. the gradient) of $\Phi_{\mathrm{T}}$ and is linear in $\Delta B_0$.

## Results

### Static magnetic fields

We first look at the dependence of the triplet yield, $\Phi_{\mathrm{T}}$, on the strength of an external static magnetic field, $B_0$, to provide a basis for the discussion of ELF magnetic field effects. *Figure 2(a)* shows $\Phi_{\mathrm{T}}$ as a function of $B_0$ in the range 0–5 mT for seven values of the radical pair lifetime, $\tau = k^{-1}$, between 30 ns and 100 µs. The salient features are as follows. (a) For lifetimes greatly in excess of 1 µs, $\Phi_{\mathrm{T}}$ is almost independent of $B_0$ and approximately equal to 0.75. (b) For intermediate lifetimes ($\tau \approx 1$ µs), the magnetic field effect is bi-phasic: a small increase in $\Phi_{\mathrm{T}}$ for $B_0 < 1$ mT is followed by a larger decrease which levels out at fields in excess of 5 mT. The initial 'bump', known as the 'low field effect' (LFE), has been observed in experiments on a variety of radical pair reactions (*Maeda et al., 2012*; *Kattnig et al., 2016a*; *Timmel et al., 1998*; *Brocklehurst, 1976*; *Eveson et al., 2000*). (c) When the lifetime of the radical pair is much shorter than 1 µs, the LFE vanishes and the magnetic field effect becomes mono-phasic. (d) Compared to $B_0 = 0$, a typical geomagnetic field ($B_0 = 50$ µT) produces the largest change in $\Phi_{\mathrm{T}}$ when the LFE is at its most prominent, that is for lifetimes, $\tau \approx 1$ µs. This can be seen more clearly in *Figure 2(b)*.

All the features of *Figure 2* that relate to Earth-strength magnetic fields have been observed experimentally and can readily be understood (*Rodgers, 2009*; *Timmel et al., 1998*). Briefly, if the radical pair exists for much less than 1 µs, there is no time for significant Larmor precession

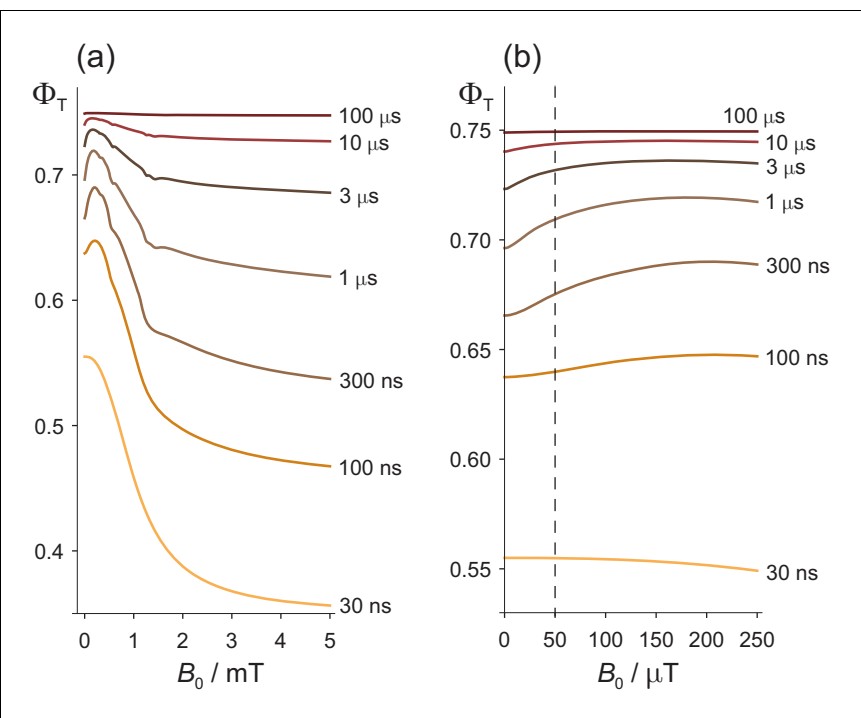

**Figure 2.** Effects of a static magnetic field on a model of the [FAD$^{\bullet-}$ TrpH$^{\bullet+}$] radical pair in cryptochrome. (a) Dependence of the reaction yield $\Phi_{\mathrm{T}}$ on the static magnetic field strength, $B_0$, for different values of the radical pair lifetime, $\tau$. The spin relaxation time is 1 µs. (b) An expanded view of the low field region of (a) with $B_0 = 50$ µT indicated by the dashed line. The difference between $\Phi_{\mathrm{T}}$ at zero field and at 50 µT is largest when $\tau \approx 1$ µs. The corresponding calculations for a spin relaxation time of 0.1 µs are shown in *Figure 2—figure supplement 1*.
DOI: https://doi.org/10.7554/eLife.44179.003
The following figure supplement is available for figure 2:

**Figure supplement 1.** $\Phi_{\mathrm{T}}(B_0)$ for $r = 10^7$ s$^{-1}$.
DOI: https://doi.org/10.7554/eLife.44179.004

(frequency, 1.4 MHz) in a 50 µT field and hence no additional singlet−triplet interconversion. If the radical pair lives much longer than 1 µs, spin relaxation destroys the spin coherence and establishes a 1:3 singlet:triplet ratio before the radicals can react. In both cases the effect of a 50 µT magnetic field effect is minimal (*Kattnig et al., 2016b*; *Kattnig et al., 2016c*; *Worster et al., 2016*). The biphasic character of $\Phi_T$ arises from two distinct mechanisms: oscillation of low frequency zero-quantum coherences at small $B_0$ and energetic isolation of two of the three triplet sub-levels at high $B_0$ (*Timmel et al., 1998*; *Till et al., 1998*). The slight irregularities in $\Phi_T$, visible in some of the traces in *Figure 2(a)*, arise from avoided energy-level crossings (*Timmel et al., 2001*; *Hiscock et al., 2016*). Such features are usually imperceptible for more realistic spin systems containing a larger number of nuclear spins.

Two other features of *Figure 2* will be relevant for our discussion of ELF effects. At $B_0 \approx 50$ µT, the gradient, $\mathrm{d}\Phi_T/\mathrm{d}B_0$, is positive and the curvature, $\mathrm{d}^2\Phi_T/\mathrm{d}B_0^2$, mostly negative (i.e. concave downward).

## ELF magnetic fields

Before showing simulations of $mfe_{ELF}$, we anticipate some of its properties. The dependence of $mfe_{ELF}$ on the curvature of $\Phi_T$ in *Equation (10)* is rationalised in *Figure 3(a)* which shows that $mfe_{ELF}$ = 0 if $\Phi_T$ depends *linearly* on $B$ in the range $(B_0 - B_1) \leq B \leq (B_0 + B_1)$. In that case, the average effect of static and ELF fields together (blue arrow) is the same as for the static field alone (green arrow). The change in $\Phi_T$ when $0 \leq \alpha \leq \frac{1}{2}\pi$ exactly cancels that for $\frac{1}{2}\pi \leq \alpha \leq \pi$. This is not the case when $\Phi_T$ has a *non-linear* dependence on $B$, *Figure 3(b)*.

From *Figures 2* and *3*, we can anticipate that for $B_0 = 50$ µT and $B_1 = 1.0$ µT, $mfe_{ELF}$ will be largest when the LFE is strongest, that is when $\tau \approx 1$ µs. *Figure 4(a)* shows that this is indeed the case. For lifetimes in the range 1 ns−1 ms, the largest change in the reaction yield caused by the ELF magnetic field is −1.2 parts per million (ppm); $mfe_{ELF}$ is negative because $\Phi_T^{(2)}(B_0) < 0$ for most values of $\tau$. The significance of the sign of $mfe_{ELF}$ is discussed in 'Signs of $mfe_{ELF}$ and $mfe_{GMF}$'.

Although magnetic field effects on purified cryptochromes arise from [FAD$^{\bullet-}$ TrpH$^{\bullet+}$], there is some evidence that, in vivo, FAD$^{\bullet-}$ is partnered by a radical with fewer, smaller hyperfine interactions than TrpH$^{\bullet+}$ (*Lee et al., 2014*; *Ritz et al., 2009*). The simulation in *Figure 4(a)* was therefore repeated for a radical pair, [FAD$^{\bullet-}$ Z$^{\bullet}$], identical to [FAD$^{\bullet-}$ TrpH$^{\bullet+}$] except that the second radical, Z$^{\bullet}$, has no hyperfine interactions. Previous studies, both experimental and theoretical, have shown that

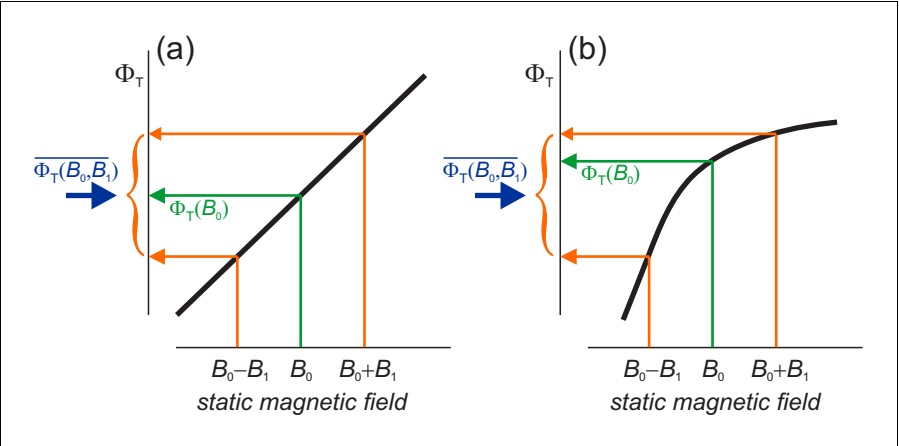

**Figure 3.** Schematic representations (thick black lines) of the dependence of the reaction yield, $\Phi_T$, on the strength of a static magnetic field. The orange arrows indicate the yields for the maximum and minimum values of $B$ in *Equation (5)*. The green arrows show the yields when $B = B_0$. The blue arrows mark the free radical yields averaged over the phase of the ELF field, *Equation (6)* . (a) When $\Phi_T$ is linear in $B$, the effect of the GMF and the ELF field *together* equals that of the GMF *alone*. (b) When $\Phi_T$ is non-linear, the effects of GMF plus ELF and GMF-alone differ. $\Phi_T$ in (b) has been drawn with a negative curvature (concave downward), with the result that $\overline{\Phi_T(B_0, B_1)} < \Phi_T(B_0)$ and $mfe_{ELF} < 0$ (*Equation (7) and (10)*). A positive curvature (concave upward) would give the opposite signs.
DOI: https://doi.org/10.7554/eLife.44179.005

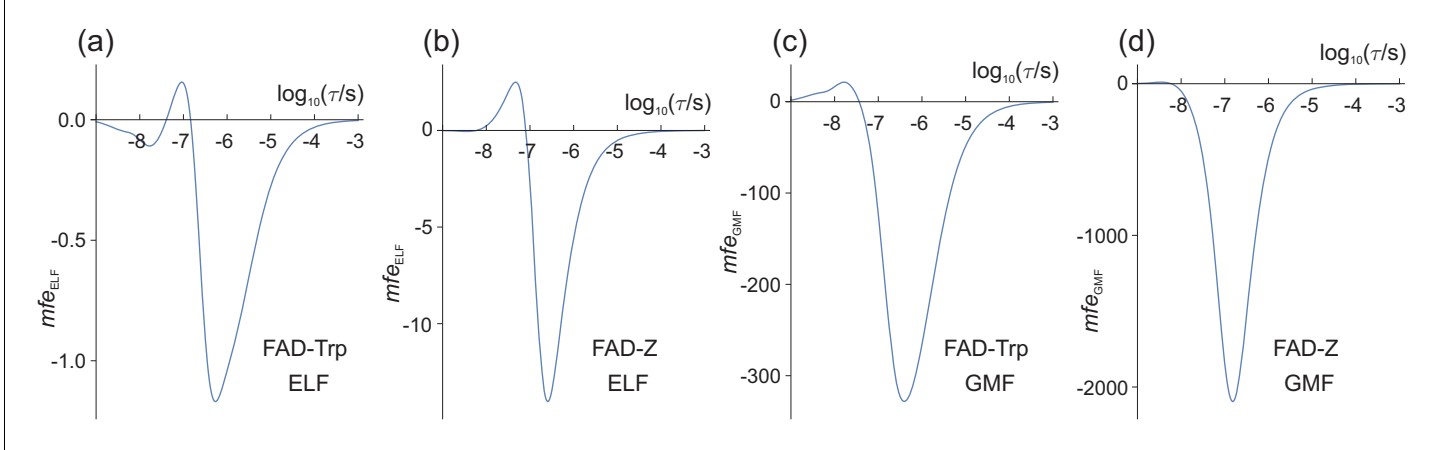

**Figure 4.** Magnetic field effects (in ppm) on model radical pairs as a function of the lifetime of the radical pair, $\tau$, in the range 1 ns−1 ms. The spin relaxation time is 1 μs. (a) and (c) A model of the [FAD$^{\bullet-}$ TrpH$^{\bullet+}$] radical pair in cryptochrome, containing two $^{14}$N nuclei in the FAD$^{\bullet-}$ radical and one in the TrpH$^{\bullet+}$ radical. (b) and (d) A model of the [FAD$^{\bullet-}$ Z$^{\bullet}$] radical pair in cryptochrome in which the Z$^{\bullet}$ radical has no hyperfine interactions. (a) and (b) $mfe_{ELF}$ (*Equation (10)*) for $B_0$ = 50 μT, $B_1$ = 1.0 μT. (c) and (d) $mfe_{GMF}$ (*Equation (12)*) for $B_0$ = 50 μT, $\Delta B_0$ = −1.0 μT. Note the different vertical scales of the four panels. The validity of the Taylor series expansion (*Equation (9)*) was confirmed by numerical integration over $\alpha$, the phase of the ELF field (*Figure 4—figure supplement 1*). Corresponding calculations for a spin relaxation time of 0.1 μs are shown in *Figure 4—figure supplement 2*. Corresponding versions of (a) and (c) when 25 μT $\leq B_0 \leq$ 65 μT are shown in *Figure 4—figure supplement 3*.

DOI: https://doi.org/10.7554/eLife.44179.006

The following figure supplements are available for figure 4:

**Figure supplement 1.** $mfe_{ELF}^{max}$ evaluated by numerical integration.
DOI: https://doi.org/10.7554/eLife.44179.007
**Figure supplement 2.** $mfe_{ELF}^{max}$ and $mfe_{GMF}^{max}$ for $r = 10^7 s^{-1}$.
DOI: https://doi.org/10.7554/eLife.44179.008
**Figure supplement 3.** $mfe_{ELF}$ and $mfe_{GMF}$ for 25 μT $\leq B_0 \leq$ 65 μT.
DOI: https://doi.org/10.7554/eLife.44179.009

such highly asymmetric radical pairs show larger low field effects than when the hyperfine interactions are more evenly distributed between the two radicals (*Lee et al., 2014*; *Stass et al., 1995*; *Rodgers et al., 2007*). As *Figure 4(b)* shows, this is also the case for ELF fields. The maximum ELF effect for the model [FAD$^{\bullet-}$ Z$^{\bullet}$] pair is −14 ppm, roughly an order of magnitude larger than the largest $mfe_{ELF}$ for [FAD$^{\bullet-}$ TrpH$^{\bullet+}$].

## Comparison of ELF and GMF effects

*Figure 4(a) and (b)* suggest that the effect of a 1 μT ELF magnetic field is likely to be no more than a few parts per million. How should such numbers be interpreted in terms of any potential biological significance? The simplest approach would be to assert that changes in the yield of a chemical reaction smaller than, say, 100 ppm (i.e. 0.01%) can be dismissed as insignificant. Although arguably reasonable, the choice of any such threshold is necessarily arbitrary, especially as we have deliberately refrained from considering specific biological processes. An alternative would be to calculate a 'signal-to-noise' ratio in which the 'signal' would be a change in concentration (e.g. of a reactive oxygen species) induced by the ELF field and the 'noise' would be the average background fluctuation in that concentration. Then, with some confidence, one could reject potential ELF effects that had a predicted signal-to-noise ratio less than 1.0. Clearly, this too is unsatisfactory due to lack of knowledge of both the 'noise' and the link between the radical pair reaction and a biological end-point. We have therefore chosen not to rely solely on the absolute values of $mfe_{ELF}$ but instead to compare predicted ELF effects with the changes in $\Phi_T$ that would arise from small variations in the strength of the geomagnetic field such as might be experienced by moving to a different point on the Earth's surface. Performing the two calculations using the same model and the same parameters allows ELF effects to be put into a more readily appreciated perspective. Furthermore, by taking the ratio of $mfe_{ELF}$ (*Equation (10)*) and $mfe_{GMF}$ (*Equation (12)*) under identical conditions, any errors arising from the approximations and assumptions of the model will tend to cancel.

To facilitate comparison with the ELF effects shown in *Figure 4(a) and (b)* (for which $B_1 = 1.0$ µT), we keep $B_0$ fixed at 50 µT and choose $\Delta B_0 = -1.0$ µT (negative, so that $mfe_{\text{GMF}}$ has, for the most part, the same sign as $mfe_{\text{ELF}}$). The results are shown in *Figure 4(c) and (d)* for the model [FAD$^{\bullet-}$ TrpH$^{\bullet+}$] and [FAD$^{\bullet-}$ Z$^{\bullet}$] radical pairs respectively. In both cases, the largest $mfe_{\text{GMF}}$ exceeds the largest $mfe_{\text{ELF}}$ by two orders of magnitude. In the following we denote these maximum magnetic field effects $mfe_{\text{GMF}}^{\text{max}}$ and $mfe_{\text{ELF}}^{\text{max}}$.

*Equation (12) and (10)* show that $mfe_{\text{GMF}}$ and $mfe_{\text{ELF}}$ scale, respectively, linearly in $\Delta B_0$ and quadratically in $B_1$. We can therefore use Figure 4 to predict $mfe_{\text{GMF}}$, $mfe_{\text{ELF}}$ and their ratio for different values of $\Delta B_0$ and $B_1$, provided $|\Delta B_0| \ll B_0$ and $B_1 \ll B_0$. For example, for Figures 4(a) and 4(c), $mfe_{\text{GMF}}^{\text{max}}$ is approximately 280 times larger than $mfe_{\text{ELF}}^{\text{max}}$. $mfe_{\text{GMF}}$ and $mfe_{\text{ELF}}$ should therefore have similar maximum amplitudes when $\Delta B_0$ is reduced by a factor of 280. Thus, for the model of [FAD$^{\bullet-}$ TrpH$^{\bullet+}$], a $-3.6$ nT change in the geomagnetic field should have roughly the same effect on $\Phi_T$ as would a 1 µT ELF field. For the simpler [FAD$^{\bullet-}$ Z$^{\bullet}$] radical pair (Figures 4(b) and 4(d)), where the ratio of $mfe_{\text{GMF}}^{\text{max}}$ to $mfe_{\text{ELF}}^{\text{max}}$ is 150), the corresponding $\Delta B_0$ would be $\approx -6.7$ nT.

## Comparison with temperature effects

A different comparison, which may also help to put the predicted ELF effects into context, relates to the daily variation in body temperature which, for a healthy adult, is typically ±0.5°C. The two parameters in our calculation that are most likely to be temperature-sensitive are the rate constants for the reactions ($k$) and the relaxation ($r$) of the radicals. By analogy with *Equation (10) and (12)*, we define $Te_{\Delta k, \Delta r}$ ($Te$ = temperature effect) as the change in $\Phi_T$ resulting from small increases ($\Delta k$ and $\Delta r$) in $k$ and $r$, respectively,

$$Te_{\Delta k, \Delta r} = \frac{\Phi_T^{(k+\Delta k, r+\Delta r)} - \Phi_T^{(k,r)}}{\Phi_T^{(k,r)}} . \tag{13}$$

To obtain crude estimates of the temperature dependence of $\Phi_T$, we assume that both $k$ and $r$ increase/decrease by 0.1% for a temperature rise/fall of 1°C. Assuming Arrhenius behaviour (rate $\propto \exp[-E_a/RT]$) and $T = 37$°C, this corresponds to a small activation energy, $E_a = 0.80$ kJ mol$^{-1}$ ($\approx 0.31$ $RT$). Larger activation energies would lead to more dramatic changes in these rate constants.

Taking $k = r = 10^6$ s$^{-1}$, $B_0 = 50$ µT and $\Delta B_0 = B_1 = 0$, we find $Te_{\Delta k, 0} = -36$ ppm and $Te_{0, \Delta r} = +21$ ppm for [FAD$^{\bullet-}$ TrpH$^{\bullet+}$] and $Te_{\Delta k, 0} = -54$ ppm and $Te_{0, \Delta r} = +27$ ppm for [FAD$^{\bullet-}$ Z$^{\bullet}$]. The two temperature effects are additive: increasing both rate constants by 0.1% simultaneously results in $Te_{\Delta k, \Delta r} = -15$ ppm for [FAD$^{\bullet-}$ TrpH$^{\bullet+}$] and $-27$ ppm for [FAD$^{\bullet-}$ Z$^{\bullet}$]. These numbers should be compared with the corresponding values of $mfe_{\text{ELF}}$ in *Figure 4(a) and (b)*.

For convenience, all the numerical results reported in this section are collected in *Table 1*, with $\Omega_{\text{ELF}}^{\text{GMF}}$ defined as the ratio $mfe_{\text{GMF}}^{\text{max}}/mfe_{\text{ELF}}^{\text{max}}$.

## Discussion

### Signs of $mfe_{\text{ELF}}$ and $mfe_{\text{GMF}}$

*Equations (10) and (12)* predict that $mfe_{\text{ELF}}$ and $mfe_{\text{GMF}}$ depend, respectively, on the curvature and gradient of $\Phi_T$ at $B_0 = 50$ µT and *Figure 2* shows that these quantities are, respectively, negative and positive for radical pairs that have lifetimes between ~100 ns and 10 µs. Our simulations (*Figure 4*) bear out this expectation. The presence of a weak ELF field or a small decrease in the strength of the geomagnetic field both reduce the yield of the triplet product and correspondingly increase $\Phi_S$, the singlet yield ($\Phi_S + \Phi_T = 1$, *Figure 1(b) and (c)*). The signs of $mfe_{\text{ELF}}$ and $mfe_{\text{GMF}}$ are both reversed if the radical pair starts out as a triplet rather than a singlet. Given our wish to be agnostic about the chemical and biological details, it makes no sense to interpret the absolute signs of $mfe_{\text{ELF}}$ and $mfe_{\text{GMF}}$.

Nor are the relative signs of $mfe_{\text{ELF}}$ and $mfe_{\text{GMF}}$ unconditionally useful for interpreting experimental data. Although $mfe_{\text{ELF}}$ and $mfe_{\text{GMF}}$ have mostly negative values for both radical pairs in *Figure 4* (where $r = 10^6$ s$^{-1}$), positive values of both quantities are predicted when $r = 10^7$ s$^{-1}$ (*Figure 4—*

**Table 1.** Changes in reaction product yields (in ppm) arising from small changes in magnetic field strengths and temperature

| | | [FAD$^{\bullet-}$ TrpH$^{\bullet+}$] | [FAD$^{\bullet-}$ Z$^{\bullet}$] |
|---|---|---|---|
| $mfe_{\mathrm{ELF}}^{\max}$* | $B_1 = 1.0~\mu T$ | −1.2 | −14 |
| $mfe_{\mathrm{GMF}}^{\max}$* | $\Delta B_0 = -1.0~\mu T$ | −330 | −2100 |
| $\Omega_{\mathrm{ELF}}^{\mathrm{GMF}}$† | | 280 | 150 |
| $Te_{\Delta k,0}$‡ | $B_1 = \Delta B_0 = 0$ | −36 | −54 |
| $Te_{0,\Delta r}$§ | $B_1 = \Delta B_0 = 0$ | +21 | +27 |
| $Te_{\Delta k,\Delta r}$¶ | $B_1 = \Delta B_0 = 0$ | −15 | −27 |

*$r = 10^6~\mathrm{s}^{-1}$, $B_0 = 50~\mu T$

†$\Omega_{\mathrm{ELF}}^{\mathrm{GMF}} = mfe_{\mathrm{GMF}}^{\max}/mfe_{\mathrm{ELF}}^{\max}$

‡$k = r = 10^6~\mathrm{s}^{-1}$, $\Delta k = 10^3~\mathrm{s}^{-1}$, $\Delta r = 0$, $B_0 = 50~\mu T$

§$k = r = 10^6~\mathrm{s}^{-1}$, $\Delta k = 0$, $\Delta r = 10^3~\mathrm{s}^{-1}$, $B_0 = 50~\mu T$

¶$k = r = 10^6~\mathrm{s}^{-1}$, $\Delta k = \Delta r = 10^3~\mathrm{s}^{-1}$, $B_0 = 50~\mu T$

DOI: https://doi.org/10.7554/eLife.44179.010

figure supplement 2) as a result of the different dependence of $\Phi_T$ on $B_0$ (Figure 2—figure supplement 1).

## Assumptions and approximations

Before attempting to draw conclusions from the data in *Table 1*, we first assess the validity of the assumptions and approximations used to obtain them. The most important of these concern (*a*) spin relaxation, (*b*) hyperfine interactions, (*c*) electron-electron interactions, (*d*) relative motion of the radicals, and (*e*) the ELF field.

(*a*) Spin relaxation of organic radicals at physiological temperatures is likely to be faster than the 1 μs we have assumed here and unlikely to be slower (*Kattnig et al., 2016b*). Although one could imagine longer relaxation times (e.g. ~10 μs) in an avian protein (e.g. cryptochrome) optimised for exquisitely sensitive detection of the geomagnetic field, it is highly improbable that these properties could arise by chance. It would require that the random thermal motions of the radicals had very low amplitude or very high frequency or both (*Hore and Mouritsen, 2016*). Neither seems probable in a biological environment at physiological temperatures. If spin relaxation is faster than 1 μs, all magnetic field effects will be smaller than those in *Table 1*. For example, $mfe_{\mathrm{ELF}}$ and $mfe_{\mathrm{GMF}}$ are both reduced by roughly an order of magnitude when $r^{-1} = 0.1$ μs, although their ratio is not greatly changed (see *Figure 4—figure supplement 2*). By contrast, changing the relaxation time from 1 μs to 10 μs increases $mfe_{\mathrm{ELF}}^{\max}$ and $mfe_{\mathrm{GMF}}^{\max}$ by factors of 2.7 and 1.7 respectively, and consequently reduces $\Omega_{\mathrm{ELF}}^{\mathrm{GMF}}$ by 40%. There are only minor changes in $mfe_{\mathrm{ELF}}^{\max}$ and $mfe_{\mathrm{GMF}}^{\max}$ for relaxation times longer than 10 μs. This is because 10 μs is already an order of magnitude longer than the reciprocal of the electron Larmor frequency (~700 ns) in a static magnetic field of 50 μT. Although increasing the relaxation time beyond 10 μs would allow the radical pair to respond more sensitively to static fields weaker than 50 μT, the magnetic field effect at 50 μT field is already at its maximum.

(*b*) Organic radicals with the properties needed to show magnetic field effects almost always have more hyperfine interactions than the one or two that have been included here (*Gerson and Huber, 2003*). The semi-occupied molecular orbital that contains the unpaired electron in an organic radical is often delocalised so that the electron interacts with several nearby hydrogen and/or nitrogen nuclei. Our experience of spin dynamics simulations has been that the more hyperfine interactions included in both radicals, the smaller the effects of weak magnetic fields. Conversely, the magnetic sensitivity is generally larger when one of the radicals has just a few small hyperfine interactions or none at all (*Lee et al., 2014*). As we have seen with [FAD$^{\bullet-}$ Z$^{\bullet}$], removing the single hyperfine interaction from the model TrpH$^{\bullet+}$ radical, increases $mfe_{\mathrm{ELF}}$ and $mfe_{\mathrm{GMF}}$ by about an order of magnitude (and reduces $\Omega_{\mathrm{ELF}}^{\mathrm{GMF}}$ by a factor of 2, from 280 to 150, *Table 1*). Biologically relevant radicals with no or very few hydrogen and nitrogen atoms near the unpaired electron are scarce. Superoxide, $O_2^{\bullet-}$, is one and nitric oxide, $NO^{\bullet}$, is another but both normally have such fast spin relaxation

(probably ~1 ns (*Karogodina et al., 2009*; *Karogodina et al., 2011*)) that they can only participate in magnetically sensitive reactions under rather special circumstances (*Hogben et al., 2009*; *Kattnig, 2017*). The only other radical discussed in the context of magnetoreception is that formed by oxidation of ascorbic acid, Asc•−. [FAD•− Asc•−] is predicted to show magnetic field effects larger than [FAD•− TrpH•+] but smaller than [FAD•− Z•] (*Kattnig et al., 2016a*; *Lee et al., 2014*; *Nielsen et al., 2017*). In short, it is difficult to imagine a biologically plausible radical pair whose hyperfine couplings make it significantly more sensitive to weak magnetic fields than the simplified model of [FAD•− TrpH•+] we have considered here.

(*c*) In all of the simulations discussed here, we have ignored the possibility that the unpaired electrons in the two radicals have a mutual spin-spin interaction. Although this has been common practice in theoretical treatments of radical pair magnetoreception (*Hore and Mouritsen, 2016*), it only starts to be a good approximation when the exchange and dipolar interactions, which tend to block singlet-triplet interconversion, are smaller than the geomagnetic field (~50 µT). This in turn would require the radicals to be separated by more than 3.8 nm (*Efimova and Hore, 2008*). At such large distances, it is extremely doubtful whether spin-selective recombination would be fast enough to compete with spin relaxation. Inclusion of realistic spin-spin interactions can easily attenuate the predicted magnetic field effects by an order of magnitude or more (*Efimova and Hore, 2008*).

(*d*) We have treated the radicals as fixed in space partly because that is the case in cryptochrome but more importantly because the effects of weak fields are more pronounced when the radicals are unable to move freely. Translationally diffusing radicals generally show stronger LFEs when their motions are restricted, for example by confinement within a micelle (*Eveson et al., 2000*; *Turro and Weed, 1983*; *Sakaguchi and Hayashi, 1984*). One of the reasons for this is that without such constraints, translational motion is an efficient source of spin relaxation via modulation of exchange interactions (*Shushin, 1991*). Another is that a fraction of the radical pairs inevitably diffuse apart without ever having the opportunity to undergo the spin-selective reaction(s) essential for a magnetic field effect.

(*e*) We have taken the ELF field to be linearly polarised and aligned parallel to the GMF such that the total field experienced by the radicals varies between 49 and 51 µT. Any other choice would result in a smaller range of total field strengths and therefore smaller ELF effects. If a 1 µT ELF field is randomly oriented with respect to a 50 µT static field, then the average total field fluctuates between 49.37 and 50.64 µT, that is a variation of ~0.64 µT instead of 1 µT. Given the quadratic dependence on the amplitude of the ELF field (*Equation (10)*), our choice of parallel fields overestimates the ELF effect by a factor of $\sim(0.64)^{-2} \approx 2.4$ compared to random orientations.

To summarise, in all five cases (*a-e*) we have deliberately designed the model and chosen its parameters in such a way as to maximise and/or overestimate the sensitivity to weak magnetic fields.

However, as we now discuss, we have also made a few assumptions that could in practice, or at least in principle, cause $mfe_{\mathrm{ELF}}$ and/or $mfe_{\mathrm{GMF}}$ to exceed the values in *Table 1*. First we deal with three factors related to the reaction scheme in *Figure 1(b)*. (*f*) Rather than starting as a singlet, the radical pair might be formed in a triplet state or arise from the encounter of radicals with uncorrelated electron spins ('F-pairs') (*Steiner and Ulrich, 1989*; *Salikhov et al., 1984*). (*g*) The singlet and triplet states of the radical pair could react with different rate constants ('asymmetric recombination') (*Lewis et al., 2014*). (*h*) Only one of the competing reaction channels needs to be spin-selective: the other can be independent of spin and proceed with identical singlet and triplet rate constants, as is the case for [FAD•− TrpH•+] in cryptochrome ('semi-spin-selective recombination') (*Maeda et al., 2012*). Simulations of the model [FAD•− TrpH•+] radical pair show that relaxing these three assumptions can increase $mfe_{\mathrm{ELF}}$ by up to a factor of 8 (*Blunsden, 2018*). For example, $mfe_{\mathrm{ELF}}^{\max}$ increases by 40% if $k_{\mathrm{S}}$ and $k_{\mathrm{T}}$ are no longer constrained to be equal but is 20% smaller for semi-spin-selective recombination. Somewhat larger changes in $mfe_{\mathrm{ELF}}^{\max}$ are found when the radical pair starts in a triplet state: an 8-fold increase for asymmetric recombination and 7-fold when only the singlet recombination step is spin-selective. However, in all cases $mfe_{\mathrm{GMF}}^{\max}$ shows similar changes such that $\Omega_{\mathrm{ELF}}^{\mathrm{GMF}}$ is no more than 20% larger than that for symmetric recombination of a singlet-born radical pair. We stress that these increases in magnetic sensitivity are maxima, corresponding to particular choices of rate constants, initial spin state and reaction scheme. Although evolution could have exploited such

conditions, for example to make a more efficient magnetic compass, it seems improbable that they would occur by chance.

(*i*) In calculating the effects of a 1 µT ELF magnetic field we have used a geomagnetic field strength (50 µT) appropriate for northern Europe. Repeating the [FAD$^{\bullet-}$ TrpH$^{\bullet+}$] simulations (*Figure 4*) with $B_0$ = 25 µT and 65 µT (the extreme values of the Earth's field) give $mfe_{\text{ELF}}$ values respectively 2.2 times bigger and 1.4 times smaller than for $B_0$ = 50 µT (see *Figure 4—figure supplement 3*). Similar effects were found for $mfe_{\text{GMF}}$ which was 1.4 times bigger when $B_0$ = 25 µT and 1.2 times smaller when $B_0$ = 65 µT. We note that Swanson and Kheifets considered the possibility that ELF magnetic fields could have different consequences at different locations on the Earth's surface due to variations in the geomagnetic field. To test this, they analysed 15 epidemiological studies and found 'some, but rather limited and not statistically significant, evidence' for a correlation between ELF exposure and incidence of childhood leukaemia (*Swanson and Kheifets, 2012*).

Over the years, various mechanisms that could amplify small radical pair magnetic field effects have been suggested, none of which we have so far considered. Four are mentioned here. First (*j*), is the possibility that a superparamagnetic nanoparticle could boost the strength of the ELF field experienced by a nearby radical pair (*Binhi, 2008*). Briefly, the idea is that the 50/60 Hz field could align the instantaneous, fluctuating magnetic moment of the nanoparticle such that the magnetic field close to its surface would be much stronger than the external ELF field but would still oscillate at 50/60 Hz. We can assess the likely importance of this effect using a thermodynamic argument based on ferritin, a naturally occurring superparamagnetic protein, which has an instantaneous magnetic moment, $m \approx 300\,\mu_{\text{B}}$ ($\mu_{\text{B}}$ is the Bohr magneton) (*Kilcoyne and Cywinski, 1995*; *Vohralik and Lam, 2009*; *Worster and Hore, 2018*). The energy of the interaction of ferritin with a $B_1$ = 1 µT magnetic field, $\sim mB_1$, is a million times smaller than the thermal energy, $k_{\text{B}}T$, at physiological temperature. The alignment of the nanoparticle's magnetic moment, and therefore the amplification of the ELF field, will consequently be negligible.

In the context of cryptochrome-based magnetoreception, two amplification mechanisms have been proposed, and in one case demonstrated experimentally. They rely on (*k*) slow radical termination reactions (*Kattnig et al., 2016a*) and (*l*) reactions of the radicals with paramagnetic scavengers (*Kattnig, 2017*; *Kattnig and Hore, 2017*). The latter mechanism has the interesting and potentially important property that it seems to be immune to fast spin relaxation in one of the radicals (*Kattnig, 2017*), opening the possibility that superoxide and other reactive oxygen species could be involved in responses to weak magnetic fields. Until the mechanism is confirmed experimentally, it is difficult to know how much amplification to expect.

The final amplification mechanism (*m*) is more speculative (giving, perhaps, greater scope for disproportionate enhancements of ELF magnetic field effects). It requires a reaction scheme involving chemical feedback in which a radical pair is one of the key reaction intermediates (*Eichwald and Walleczek, 1996*; *Eichwald and Walleczek, 1997*; *Eichwald and Walleczek, 1998*; *Møller and Olsen, 1999*; *Møller et al., 2000*; *Møller and Olsen, 2000*; *Purtov, 2004*). The inherent non-linearity of such reactions could at least in principle allow small magnetically induced changes in radical pair lifetimes to have a disproportionately large effect on, for example, the amplitude of chemical oscillations. Further, one could imagine an oscillating reaction with a cycle time that coincidentally matched the period of the ELF field which might then be able to 'pump' the oscillations to higher amplitudes. However, this would require some degree of phase-coherence between the ELF field and the biochemical oscillator. Unless there is a mechanism by which an environmental ELF field could *entrain* a biological oscillator, coherent pumping and therefore selective amplification of 50/60 Hz magnetic field effects would seem rather unlikely. It is difficult to think of any mechanism which could explain a specific and disproportionate response of a radical pair reaction to an ELF field.

To sum up, we have discussed 13 assumptions and approximations in the simulations on which this article is based. In our judgement, the first five (*a-e*) are the most important and almost certainly lead to overestimates of the effects of both ELF and static fields. The next three (*f-h*) could, if operating simultaneously, enhance $mfe_{\text{ELF}}$ by up to an order of magnitude, but only for specific combinations of rate constants. Using a value other than 50 µT for the Earth's magnetic field (*i*) would increase or decrease $mfe_{\text{ELF}}$ and $mfe_{\text{GMF}}$, but only by a factor of ~2. We judge the next three (*j-l*) unlikely to have any substantial consequences for biological radical pair reactions that are fortuitously sensitivity to magnetic fields. All of (*a-l*) should increase or decrease $mfe_{\text{ELF}}^{\max}$ and $mfe_{\text{GMF}}^{\max}$ to

approximately the same degree making their ratio, $\Omega_{\text{ELF}}^{\text{GMF}}$, much less sensitive to the exact conditions of the simulations. Finally, it is difficult to assess the importance of amplification via chemical feedback (*m*) given the highly speculative nature of this mechanism. Certainly, a set of coupled reactions would require some very unusual (and unknown) properties to be able to boost magnetic field effects preferentially at 50/60 Hz. It is not inconceivable that there exists in human biology systems at the cellular level that have evolved sensitivity to the Earth's magnetic field. If so, then some of the above assumptions may be less secure than presented leading to underestimates of magnetic field effects. However, it is still difficult to imagine situations that would selectively enhance responses at 50/60 Hz.

## Additional aspects

Before concluding, we want to mention briefly five additional aspects of radical pair chemistry/physics.

First, when a radical pair is formed in a singlet state, but not when it is formed as an unpolarised triplet, its electron spins are entangled (*Hogben et al., 2012*). The possibility that entanglement arises naturally in the 'warm, wet and noisy' milieu of a living cell has attracted a certain amount of attention from theoretical physicists (*Cai et al., 2010*; *Gauger et al., 2011*; *Kominis, 2012*; *Pauls et al., 2013*; *Tiersch et al., 2014*; *Zhang et al., 2014*). Nevertheless, there is no apparent way in which this entanglement can enhance magnetic responses (*Hore and Mouritsen, 2016*).

Second, humans have only been exposed to widespread anthropogenic ELF fields since the early days of electrification in the late 19th century. If there is a mechanism by which radical pair reactions can be unusually sensitive to 50/60 Hz magnetic fields, it cannot have evolved in only ~100 years and would have to be a chance consequence of, for example, an unknown cellular response to the Earth's static magnetic field that had evolved at a much earlier stage in human development.

Third, radical pairs are well known to show resonant responses to radiofrequency magnetic fields in the frequency range 1–100 MHz (*Henbest et al., 2004*; *Frankevich and Kubarev, 1982*; *Woodward et al., 2001*). Similar effects cannot occur at ELF frequencies because spin relaxation will destroy all spin coherence on a timescale much faster than the 20 ms period of a 50 Hz oscillation. To put it another way: with a ~1 μs coherence lifetime, any resonance would be ~1 MHz wide, thus precluding any possibility that the sensitivity to a 50 Hz field could be greater than that for a static field.

Fourth, it appears that radical pairs can only be formed in cryptochromes when the FAD cofactor is correctly bound. Without the flavin, there is very limited scope for the intra-protein electron transfer reactions that could produce magnetically responsive radical pairs. Current evidence suggests that Type 2 vertebrate cryptochromes may be 'vestigial' flavoproteins, unable to bind FAD (*Kutta et al., 2017*). If true, then human cryptochromes (exclusively Type 2) are unlikely to form radical pairs and therefore improbable as agents of biological radical pair effects. By contrast, Type 4 cryptochromes, which are found in birds, fish and reptiles (*Kobayashi et al., 2000*; *Kubo et al., 2006*; *Kubo et al., 2010*), do bind FAD (*Ozturk et al., 2009*; *Mitsui et al., 2015*; *Qin et al., 2016*; *Wang et al., 2018*) and appear to be fit for purpose as magnetoreceptors in migratory songbirds (*Günther et al., 2018*).

Fifth, not all radical pair reactions are magnetically sensitive. For example, several flavin-dependent enzymes known or thought to proceed via radical pair intermediates show no magnetic field effects because catalysis is not rate-limited by spin-selective reaction steps (*Messiha et al., 2015*).

## Conclusions

We believe the values of $mfe_{\text{ELF}}^{\max}$ in *Table 1* (−1.2 ppm and −14 ppm for the two model radical pairs considered) provide a good estimate of the maximum fortuitous effect of a 1 μT 50/60 Hz magnetic field on *any* radical pair reaction. Given the arguments in 'Assumptions and approximations' and the scarcity of biological radicals devoid of significant hyperfine interactions, we propose that 10 ppm should be seen as an upper limit on the change in the yield of a radical pair reaction produced by a 1 μT ELF field in the presence of the Earth's magnetic field (25–65 μT).

To put this number in context, we have estimated the changes in the yields of radical pair reactions that might result from ±0.5°C variations in temperature. These changes (*Table 1*) are of the same order of magnitude (~10 ppm) as, or larger than, $mfe_{\text{ELF}}^{\max}$.

*Table 1* also contains estimates of the maximum effect ($mfe_{\mathrm{GMF}}^{\max}$) of a 1 µT reduction in the strength of the geomagnetic field in the absence of an ELF field (−330 ppm and −2100 ppm). For similar reasons, we believe these values should also be appropriate for radical pair reactions that have not been optimised for a role in magnetic sensing. The ratios of the two maximum magnetic field effects, $\Omega_{\mathrm{ELF}}^{\mathrm{GMF}} = mfe_{\mathrm{GMF}}^{\max}/mfe_{\mathrm{ELF}}^{\max}$, in *Table 1* (280 and 150 for the two model radical pairs) suggest that $\Omega_{\mathrm{ELF}}^{\mathrm{GMF}}$ is likely to be in the range 100–500 for any radical pair reaction. In other words, a 1 µT decrease in the strength of the geomagnetic field should have an effect 100–500 times larger than would a 1 µT ELF field in the presence of the geomagnetic field. Or, equivalently, a 2–10 nT change in the strength of the geomagnetic field should have a similar effect to that of a 1 µT ELF field.

The last result can be put into context by considering the variation in the strength of the geomagnetic field over the surface of the Earth: ~65 µT at the magnetic poles and ~25 µT at the magnetic equator. Given the circumference of the Earth (~40,000 km), the average variation in the geomagnetic field along a line of longitude is roughly 4 nT km$^{-1}$. Therefore, the maximum effect of a 1 µT ELF field on a radical pair reaction (10 ppm) should be comparable to the maximum effect of travelling 0.5–2.5 km along a north-south axis.

A related comparison, on a much smaller length scale, is provided by measurements of the local distortions in the Earth's magnetic field in homes in the UK caused by the proximity of ferrous objects (*Swanson, 1994*). Variations in static field strength of order 1 µT can be experienced by movement from room to room, corresponding to much larger effects on radical pair reactions than could be expected from a 1 µT ELF field.

It may also be relevant to note that the Earth's magnetic field has a component of amplitude 25–50 nT that oscillates with a 24 hr period (caused by the tidal effect of the sun's gravity on the Earth's atmosphere (*Liboff, 2014*)). Potentially, therefore, the natural diurnal variation in the geomagnetic field could have a larger effect on radical pair chemistry than a 1 µT 50/60 Hz field.

To conclude, the predicted effects of 1 µT ELF magnetic fields in the presence of the Earth's magnetic field are small (<10 ppm) and similar to or smaller than effects on the same reactions resulting either from travelling a few kilometres or from natural fluctuations in body temperature neither of which would normally be considered as potentially harmful to human health.

Implicit in everything, we have written here is the assumption that the current theory of the radical pair mechanism is complete. We cannot exclude the possibility that, despite more than 40 years of research, there is some completely unknown aspect of radical pair spin chemistry that confers a disproportionate sensitivity to ELF magnetic fields. We cannot begin to imagine what that aspect might be except to note a possibly related observation. Migratory birds are prevented from using their magnetic compass by extraordinarily weak broadband radiofrequency noise (*Ritz et al., 2009*; *Ritz et al., 2004*; *Thalau et al., 2005*; *Winklhofer et al., 2013*; *Engels et al., 2014*; *Kavokin et al., 2014*; *Wiltschko et al., 2015*; *Schwarze et al., 2016*) the predicted effects of which are far too small to be consistent with our current understanding of radical pair spin dynamics (*Kavokin, 2009*; *Hiscock et al., 2017*). It is possible, therefore, that a deeper understanding of the mechanism of avian magnetoreception will bring new insights into the risks associated with exposure to weak environmental 50/60 Hz magnetic fields.

## Acknowledgements

I am grateful to Jukka Juutilainen, Jim Metcalfe, and John Swanson for helpful comments on the manuscript and to the Electromagnetic Fields Biological Research Trust and its Scientific Advisory Committee for financial support and stimulating discussions over a period of many years.

## Additional information

### Funding

The funders had no role in study design, data collection and interpretation, or the decision to submit the work for publication.

## Author contributions
PJ Hore, conducted all aspects of the work, from conceptualisation through to writing, and revised the final article

## Author ORCIDs
PJ Hore ⓘD https://orcid.org/0000-0002-8863-570X

## Decision letter and Author response
Decision letter https://doi.org/10.7554/eLife.44179.015
Author response https://doi.org/10.7554/eLife.44179.016

# Additional files

## Supplementary files
• Source code 1. The *Mathematica* code used to calculate *Figures 2* and *4*.
DOI: https://doi.org/10.7554/eLife.44179.011

• Transparent reporting form
DOI: https://doi.org/10.7554/eLife.44179.012

## Data availability
All data generated during this study are included in the manuscript and supporting files. Code files have been provided for Figures 2 and 4.

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
