## [Decision Letter]

Thank you for submitting your article "Upper bound on the biological effects of 50/60 Hz magnetic fields mediated by radical pairs" for consideration by *eLife*. Your article has been reviewed by Arup Chakraborty as the Senior Editor, a guest Reviewing Editor, and three reviewers. The reviewers have opted to remain anonymous.

We are pleased to inform you that your article will be accepted for publication in *eLife*. We encourage you to address the remaining minor points brought up by the reviewers first.

Your paper investigates the potential effect of extremely low frequency magnetic fields on radical pair mediated chemical reactions. The radical pair mechanism is considered the most plausible chemical basis for magnetic-field-induced carcinogenicity in vivo. This suggestion is motivated by experimental and theoretical evidence that the radical pair mechanism underlies magneto reception in birds. The related public health hazard is difficult to assess because, although effects on an individual or a group of people may be small, nearly everyone in the developed world is constantly exposed to low frequency low intensity magnetic fields. The paper presents a theoretical investigation of two model radical pair mediated chemical reactions, and shows that predicted effects of low-frequency magnetic fields are likely to be similar to the effects of experiencing a small location-dependent variation in the Earth's field. The work was reviewed by 3 independent experts in the field, who make overwhelmingly positive comments.

*Reviewer #1:*

Based on our current understanding, the radical pair mechanism is the most plausible biophysical means by which a weak magnetic field (MF) might impact a biological system. Owing to the ubiquity of man-made MF in our environment, whether or not exposure to such MF therefore leads to adverse health outcomes is an important question. This manuscript presents calculations on a model radical pair system that estimate the magnitude of magnetically-induced changes based on reasonable exposure parameters. The arguments are set out very clearly and the generalisations made are reasonable. As such, this manuscript should appeal to the broad readership of *eLife*. I therefore recommend publication subject to satisfactory responses to the relatively minor points listed below.

1) Subsection “Outline” – "In cryptochrome, the singlet channel is a reverse electron transfer reaction that returns the charge-separated radical pair state to the diamagnetic ground state of the protein. The triplet channel is considered to produce the signalling state that initiates magnetic signal-transduction (24)."

There is no reason why the 'forward' reaction from the singlet radical pair state shouldn't also produce the signalling state of cryptochrome. Although I think this fact is eluded to in assumption (h) in subsection “Assumptions and approximations”, the author does so very briefly. This point might therefore be missed by the general reader who less familiar with the reaction dynamics of this system. I therefore suggest the author states this fact more explicitly and elaborates more directly on what effect, if any, this additional reaction channel would have on their reported calculations.

2) Subsection “Assumptions and approximations” – "Biologically relevant radicals with no or very few hydrogen and nitrogen atoms near the unpaired electron are scarce. Superoxide, O2•-, is one and nitric oxide, NO•, is another but both have such fast spin relaxation (probably ~1 ns (64, 65)) that they can be excluded (66)."

I think this statement is perhaps a bit too dismissive as written, especially in light of Kattnig, 2017. Although the author does mention this paper in Subsection “Assumptions and approximations” in the context of possible amplification mechanisms, they should also do so specifically in the context of possible 'resilience' to spin relaxation. They also state in Subsection “Assumptions and approximations”that "…it seems improbable that such mechanisms would lead to strong amplification by chance." 1 or 2 lines further substantiating this statement would help the general reader.

3) Subsection “Assumptions and approximations” – "We note that Swanson and Kheifets considered whether the Earth's magnetic field could be an 'effect modifier' of the correlation between ELF exposure and incidence of childhood leukaemia (74)." Please add a brief explanation about precisely what Swanson and Kheifets mean by 'effect modifier'. It would help the reader better understand its relevance to the context of the discussion.

4) In support of the author's conclusions, it might also be worth mentioning that in enzymes either known or thought to proceed via radical pair intermediates, there appears to be a pattern that catalysis is not rate limited by spin-selective chemistry, as illustrated by Messiha et al., 2014.

*Reviewer #2:*

This manuscript by Prof. Hore considers the dynamics of a simplified radical pair to estimate the maximally plausible effect of a weak 50 Hz magnetic ac field on the radical pair reaction ratio. The manuscript is very well written, motivated and provides a thorough discussion of any implicit assumptions. It is technically sound and I believe the results to be fully correct.

The manuscript comprehensively and convincingly makes the case that there is no plausible explanation of a biological reaction to prolonged ELF exposure via magnetic field effects via a radical pair mechanism. This is based on our current understanding of the properties of bio-relevant radical pairs (coming with the caveat that there remain open questions about the radical pair properties and mechanism believed to underlie the compass of certain species of songbird).

Of course, this still leaves the possibility of non-radical-pair magnetic and/or electric field effects as possible avenues by which ELFs might have biological implications. The core result of this manuscript is a 'negative one' in the sense that it effectively rules out the radical pair model as responsible for ELF effects in a biological settings, however, this is certainly a very important contribution that will support the need for looking in different places.

In summary, I consider this to be an important rigorous piece of work which can be published as is. I am not terribly familiar with the remit and criteria for publication in *eLife*, but to me this seems to be the right home of for this manuscript and I am therefore more than happy to support publication.

As a minor optional comment: I was wondering to what extent (if at all) any of the conclusions change when a longer spin relaxation time is considered? Even if this seems highly implausible, how about considering 1 µs, or possibly even a spin relaxation time as long as the radical pair lifetime?

*Reviewer #3:*

So far, the radical pair mechanism (RPM) represents the theoretically best understood candidate of a biophysical mechanism capable of explaining biological magnetic field effects caused by the earth magnetic field. For this reason, it is also of interest to understand its potential sensitivity to environmental low frequence electromagnetic (ELF) fields. A first basic theoretical study of this problem has been given by Adair in 1999. Since that time, a lot of experimental evidence and theoretical work has been accumulated regarding the magnetoreception in avian birds which emphasizes the role of the radical pair mechanism involving cryptochrome and tryptophane as radical pair forming partners in the bird's retina. Representing a functional response that has probably been optimized by evolution, investigating this specific model may serve as a good reference to assess the upper limits of sensitivity of biological radical pair systems in general. This is the task undertaken in the present paper, employing rigorous state of the art theory to quantify the effects of 1 µT ELF fields or of corresponding changes of the geomagnetic field, in addition to the effects of small temperature changes of +/-0.5 oC. It is found that the ELF effect is of the same order or smaller than changes due to ubiquitous temperature or geomagnetic field variations neither of which is considered as potentially harmful to human health.

The assumptions and approximations made in the derivation of the results are diligently discussed in great detail. The work includes 117 references, 31 involving the present author, and is an excellent and comprehensive up-to-date review of the recent literature in the field. Regarding the validity of his conclusion the author is honest enough to add the caveat that future results of scientific research (e.g. an issue still awaiting a concise theoretical explanation results from experimental findings of MHz field resonance effects disturbing the avian magnetic compass) might lead to more stringent conclusions.

The paper is of outstanding quality and should be published as it is.

---

## [Author Response]

Reviewer #1:

Based on our current understanding, the radical pair mechanism is the most plausible biophysical means by which a weak magnetic field (MF) might impact a biological system. Owing to the ubiquity of man-made MF in our environment, whether or not exposure to such MF therefore leads to adverse health outcomes is an important question. This manuscript presents calculations on a model radical pair system that estimate the magnitude of magnetically-induced changes based on reasonable exposure parameters. The arguments are set out very clearly and the generalisations made are reasonable. As such, this manuscript should appeal to the broad readership of eLife. I therefore recommend publication subject to satisfactory responses to the relatively minor points listed below.

Thank you for the helpful suggestions.

1) Subsection “Outline” – "In cryptochrome, the singlet channel is a reverse electron transfer reaction that returns the charge-separated radical pair state to the diamagnetic ground state of the protein. The triplet channel is considered to produce the signalling state that initiates magnetic signal-transduction (24)."There is no reason why the 'forward' reaction from the singlet radical pair state shouldn't also produce the signalling state of cryptochrome. Although I think this fact is eluded to in assumption (h) in subsection “Assumptions and approximations”, the author does so very briefly. T0is point might therefore be missed by the general reader who less familiar with the reaction dynamics of this system. I therefore suggest the author states this fact more explicitly and elaborates more directly on what effect, if any, this additional reaction channel would have on their reported calculations.

Good point. If the singlet *and* triplet states of the radical pair proceed to the signalling state, instead of just the triplet, the maximum effect of a 1 μT ELF magnetic field is reduced by 20% for [FAD^•−^ TrpH^•+^] (with *r* = 10^6^ s^−1^) in a 50 μT static field. This point is now dealt with in the subsection “Assumptions and approximations” together with some more detail on assumptions (*f*) and (*g*).

2) Subsection “Assumptions and approximations” – "Biologically relevant radicals with no or very few hydrogen and nitrogen atoms near the unpaired electron are scarce. Superoxide, O2•-, is one and nitric oxide, NO•, is another but both have such fast spin relaxation (probably ~1 ns (64, 65)) that they can be excluded (66)."I think this statement is perhaps a bit too dismissive as written, especially in light of Kattnig, 2017. Although the author does mention this paper in Subsection “Assumptions and approximations” in the context of possible amplification mechanisms, they should also do so specifically in the context of possible 'resilience' to spin relaxation. They also state in Subsection “Assumptions and approximations”that "…it seems improbable that such mechanisms would lead to strong amplification by chance." 1 or 2 lines further substantiating this statement would help the general reader.

Agreed. In retrospect, I think “can be excluded” and “it seems improbable that” are too categorical: both have been removed. Changes have been made in two places in the subsection “Assumptions and approximations”.

3). Subsection “Assumptions and approximations” – "We note that Swanson and Kheifets considered whether the Earth's magnetic field could be an 'effect modifier' of the correlation between ELF exposure and incidence of childhood leukaemia (74)." Please add a brief explanation about precisely what Swanson and Kheifets mean by 'effect modifier'. It would help the reader better understand its relevance to the context of the discussion.

Swanson and Kheifets used the term ‘effect modifier’ for the possibility that ELF magnetic fields could have different consequences at different locations on the Earth’s surface due to variations in the geomagnetic field. To test this, they analysed 15 epidemiological studies of power-frequency magnetic fields and childhood leukaemia and found ‘some, but rather limited and not statistically significant, evidence’. This point has been clarified in the subsection “Assumptions and approximations”.

4) In support of the author's conclusions, it might also be worth mentioning that in enzymes either known or thought to proceed via radical pair intermediates, there appears to be a pattern that catalysis is not rate limited by spin-selective chemistry, as illustrated by Messiha et al. 2014.

Good point. This is now included in the subsection “Additional aspects”.

Reviewer #2:[…] As a minor optional comment: I was wondering to what extent (if at all) any of the conclusions change when a longer spin relaxation time is considered? Even if this seems highly implausible, how about considering 1 µs, or possibly even a spin relaxation time as long as the radical pair lifetime?

I think you mean 1 ms – we considered 1 μs extensively in the manuscript. The following table gives the results of increasing the relaxation time from 1 μs to 10 μs (the first row of numbers comes from Table 1):

**Author response table 1. resptable1:** 

	[FAD^•−^ TrpH^•+^]	[FAD^•−^ Z^•^]	[FAD^•−^ TrpH^•+^]	[FAD^•−^ Z^•^]
*r* / s^−1^				
1 μs	−1.2	−14	−330	−2100
10 μs	−3.2	−18	−550	−2500